# Flux-Independent NMDAR Signaling: Molecular Mediators, Cellular Functions, and Complexities

**DOI:** 10.3390/ijms19123800

**Published:** 2018-11-29

**Authors:** Pavel Montes de Oca Balderas

**Affiliations:** 1Departamento de Neurociencia Cognitiva, Instituto de Fisiología Celular, UNAM. Av. Universidad 3000, C.U. Coyoacán, Ciudad de México. C.P. 04510, Mexico; pavel73@hotmail.com; Tel.: +52-55-5622-5623; 2Unidad de Neurobiología Dinámica, Departamento de Neuroquímica, INNN. Av. Insurgentes Sur #3877 Col. La Fama, Ciudad de México. C.P. 14269, Mexico

**Keywords:** NMDAR, ionotropic, flux-independent, signaling, calcium, LTD, endocytosis, neuroprotection, astrocyte

## Abstract

The glutamate (Glu) N-methyl-d-aspartate (NMDA) receptor (NMDAR) plays a critical role in synaptic communication given mainly by its ionotropic function that permeates Ca^2+^. This in turn activates calmodulin that triggers CaMKII, MAPK, CREB, and PI3K pathways, among others. However, NMDAR signaling is more complex. In the last two decades several groups have shown that the NMDAR also elicits flux-independent signaling (f-iNMDARs). It has been demonstrated that agonist (Glu or NMDA) or co-agonist (Glycine or d-Serine) bindings initiate intracellular events, including conformational changes, exchange of molecular interactions, release of second messengers, and signal transduction, that result in different cellular events including endocytosis, LTD, cell death, and neuroprotection, among others. Interestingly, f-iNMDARs has also been observed in pathological conditions and non-neuronal cells. Here, the molecular and cellular events elicited by these flux-independent actions (non-canonical or metabotropic-like) of the NMDAR are reviewed. Considering the NMDAR complexity, it is possible that these flux-independent events have a more relevant role in intracellular signaling that has been disregarded for decades. Moreover, considering the wide expression and function of the NMDAR in non-neuronal cells and other tissues beyond the nervous system and some evolutionary traits, f-iNMDARs calls for a reconsideration of how we understand the biology of this complex receptor.

## 1. Introduction

The glutamate (Glu) N-methyl-d-aspartate (NMDA) receptor (NMDAR) has been classically conceived as an ionotropic channel with a central role in neuronal communication, mediating synaptic plasticity mechanisms such as long-term potentiation and depression (LTP and LTD), and it is also involved in memory and learning, among other functions of the central nervous system (CNS). The NMDAR function and its Ca2+ influx has been extensively studied in the context of the synapse, along with the intracellular (IC) signaling mediated mainly by calmodulin that synergizes with other molecular mediators and activates calcium-/calmodulin-dependent protein kinase (CaMK), mitogen-activated protein kinase (MAPK), cAMP response element binding protein (CREB), and phosphatidylinositol 3-kinase (PI3K) pathways, among others [1,2,3]. Nevertheless, the IC signaling of the NMDAR is more complex, because for instance it has been found that synaptic and extra-synaptic signaling elicits opposing actions through the activation of different IC pathways that are intermingled, although some reports have not confirmed this observation. Additionally, according to the subunits assembled into the NMDAR, it elicits specific cellular responses [4,5]. Taken together, this evidence has demonstrated that the NMDAR is a complex receptor. Moreover, different groups since the 1990s have documented the flux-independent NMDAR signaling (f-iNMDARs) of this receptor, initially involved in LTD. Some of these observations suggested the release of Ca2+ from IC pools; however, this was explained as a result of Ca2+-induced Ca2+ release (CICR) elicited by the flow of Ca2+ through the NMDAR. In the 2000s, convincing studies demonstrated that ligand binding to the NMDAR without ion flux initiated IC events that lead to different cellular responses. More recently, different independent groups have gathered evidence indicating that f-iNMDARs mediates LTD and synaptic depression in response to amyloid β (Aβ). It has also been observed that these metabotropic-like, flux-independent, or non-canonical functions of the NMDAR mediate responses in cultured astrocytes (see section below).

In this review, the mechanisms and pathways involved in f-iNMDARs described in just a few dozen studies accumulated over almost 30 years are summarized. Though most of these studies have been performed in the context of the synaptic NMDAR and very few in non-neuronal cells, the aim of this review is to put them under the light of cell biology, rather than focusing mainly on the mechanisms of synaptic communication, plasticity, or neuronal electrophysiology. The reader can refer to recent reviews where the analysis is centred on these topics [6,7,8,9]. It must be noted that the NMDAR is widely expressed in other non-neuronal cells from the CNS and in cells from other tissues and origins, confirming that the NMDAR possesses regulatory mechanisms, IC signaling, and/or functions that differ from the synaptic NMDAR version [10]. In line with this thinking, it is then feasible that f-iNMDARs could have a more relevant role than acknowledged. This possibility is supported by the diversity of subunits that may be assembled into NMDARs and the molecular partners associated with them that are expressed by different cells. This exciting f-iNMDARs calls for a reconsideration of how we understand the biology of this receptor. In the following section, the essentials of the NMDAR structure and function are outlined to set up a fundamental framework for those readers that are not familiar with this receptor. The reader can refer to the early reviews where these topics have been reviewed in depth [1,2,11].

### NMDAR Essentials

The NMDAR is fundamental within the CNS since its ionotropic function mediates synaptic neuronal communication. It requires a co-agonist (Gly or d-Ser) for channel opening and is regulated by ions (Mg2+, Zn2+, H+) or other molecules. It plays a critical role in different mechanisms including memory, learning, LTP, and LTD, among others [1,2]. This role is given mainly by its location in the postsynaptic membrane, enabling the efflux of K+ and extracellular (EC) Ca2+ and Na+ influx into the postsynaptic neuron, although recent evidence also supports its presynaptic location and activity. The NMDAR does not conduct at neuronal resting membrane potential (−70 mV), because its pore is blocked by an Mg2+ ion that is retired when the postsynaptic membrane is depolarized. This occurs after the presynaptic neuron releases vesicular Glu into the synaptic cleft that in turn activates AMPA and kainate ionotropic receptors. Then, together with Glu and co-agonist binding, the NMDAR pore opens with the consequent cationic flux. This is why the neuronal NMDAR is considered a coincidence detector that requires membrane depolarization and agonist and co-agonist binding [1,2,11].

The NMDAR is assembled as a tetramer conformed by two obligate subunits GluN1 coupled to GluN2 and/or GluN3 subunits that take the other two positions (Figure 1). GluN1 has only one gene (Grin1). This subunit is present in all NMDARs described so far, because it is critical for NMDAR assembly in the endoplasmic reticulum (ER) and its IC traffic. GluN1 regulates NMDAR exit from the ER because it has ER retention signals that are masked after its assembly with other subunits that also contain ER retention signals. On the other hand, there are four GluN2 (A–D) subunits with one gene each (*Grin2a–d*) and two GluN3 subunits (A–B) also with one gene each (*Grin3a–b*). The diversity of subunits enables the assembly of different NMDARs depending on the subunits expressed by the cell. This results in NMDARs with different features in terms of regulation, IC transport, location, and biophysical properties. This is because each subunit has amino acid (aa) sequences that provide common and distinctive molecular partners as well as specific intramolecular interactions. Some of the mRNA for these subunits undergo alternative post-transcriptional splicing that results in molecular variants that confer specific functional features to the NMDAR. Additionally, the NMDAR subunits may be a target of post-translational modifications that also generate NMDAR functional variants or regulate its function. The post-translational modifications are diverse and include phosphorylation, myristilation, and proteolytic cleavage, among others. Importantly, the expression of NMDAR subunits in the CNS is regionalized and temporally regulated. Interestingly, the NMDAR is expressed by non-neuronal cells within the CNS, but it is also widely expressed in other cells and tissues including skin, leukocytes, testis, blood, the kidneys, vasculature, and the pancreas, among many others [1,2,10].

All NMDAR subunits share a common structure with an N-terminal EC region of ≈500 aa, a trans membrane domain with three transmembrane helices (M1, M3, and M4) and an IC loop, or M2, which re-enters the membrane. All M segments are involved in pore opening. M2 lines the IC portion of the ion channel pore, whereas M3 forms the EC region of the pore. NMDAR subunits also contain an EC loop that interacts with the ligand and an IC C-terminal domain that ranges from ≈200 aa in GluN1 and GluN3 to ≈500 aa in GluN2. The terminal EC region is comprised by two functional domains: the N-terminal domain (NTD) involved in subunit–subunit molecular interactions and NMDAR modulation and the ligand binding domain (LBD), which, in close interaction with the EC loop, shapes the ligand binding site: Glu (or the aa derivative NMDA) for GluN2 and Gly or d-Ser for GluN1 and GluN3. The IC C-terminal domain is the main region mediating NMDAR molecular interactions and thus regulates its functional properties, for instance, in the synapse, with the post-synaptic density (PSD) proteins that mediate the assembly of molecular clusters (Figure 1) [1,11].

The NMDAR is a cationic channel with similar permeability to Na+ and K+ but a higher permeability to Ca2+ that depends upon the GluN2 subunit (P_Ca_/P_X_ = 1.8–4.5). This higher permeability to Ca2+ has been related to a Ca2+ binding site located at the Q/R/N site in the apex of M2 and to the DRPEER motif in GluN1 that also binds Ca2+ and is located at the external entrance to the ion channel, that directly contacts the pore wall [1,11]. Ca2+ entry through the NMDAR activates IC signaling pathways that apparently depend upon the NMDAR synaptic or extra-synaptic location and the GluN2 subunit assembled (see below) [4]. These pathways are involved in neuronal survival, growth, and differentiation, among other functions. On the other hand, it is also known that the excessive and persistent activation of the NMDAR results in the mechanisms known as excitotoxicity, elicited by the excess of IC Ca2+ (iCa2+) with the consequent activation of IC pathways that lead to neuronal death (see below) [12]. Importantly, the Ca2+-dependent IC pathways activated by the NMDAR have also been referred to as Ca2+ flux-dependent metabotropic signaling [13] that should not be confounded with the flux-independent signaling, which is the subject of this review.

Despite the NMDAR wide expression and distribution in the cells and tissues of mammals [10], most research has been done mainly in the neuronal-synaptic context as a cationic channel [1,2]. Nonetheless, the NMDAR is expressed by other cells of the CNS, such as astrocytes or oligodendroglia, and in non-CNS cells such as endothelium, platelets, and lymphocytes, among others, in which its role is poorly studied [10]. Furthermore, there are a few dozen reports that have demonstrated Ca2+ f-iNMDARs and functions, but only some have explored the molecular mechanisms that underpin this function. This pool of references is reviewed in the following sections.

## 2. Approaches to Study f-iNMDARs

In this section, the approaches that have been employed to study f-iNMDARs are synthesized. The aim of this effort is to obtain a comprehensive picture of them to further establish new experimental paradigms that allow the dissection of ionotropic and f-iNMDARs. Considering the new findings reviewed here that demonstrate the flux-independent function of this Glu receptor, it is required that these novel signaling mechanisms are not further disregarded, if we intend to advance in the understanding on this complex molecule and its functions in the CNS and other cells and tissues beyond it.

The main approach is the use of the Glu-site-competitive inhibitor (2R)-amino-5-phosphonovaleric acid (APV or AP5) and the non-competitive, irreversible pore blocker (5S,10R)-(+)-5-methyl-10,11-dihydro-5H-dibenzo[a,d]cyclohepten-5,10-imine hydrogen maleate (dizocilpine or MK-801). When contrasting the effects of these inhibitors, it is possible to isolate whether ligand binding has f-iNMDARs, although some controversies exist because MK-801 requires the channel to be open, thus making it possible that some Ca2+ flows. Therefore, MK-801 has been used together with other approaches to reduce this possibility. Other inhibitors of the Glu binding site have been used, such as 4-(3-phosphonopropyl) piperazine-2-carboxylic acid (CPP). Additionally, the specific GluN2B inhibitors ifenprodil and Ro 256981 were employed when the role of the NMDAR containing GluN2B subunit (NMDAR2B) was tested. In addition, since it has been demonstrated that the NMDAR channel opening is achieved only after agonist and co-agonist binding, the use of Gly-site-competitive inhibitors allows one to analyze IC events in response to the Glu site ligand binding without ion flux. These inhibitors are 7-chlorokynurenic acid (7CK), 5,7-dichlorokynurenic acid (5,7DCK), L689560, or CGP 78708. Under the same rationale, other experimenters have included the use of Gly or Ser degrading enzymes. Additional approaches include the manipulation of ionic EC concentrations. Since the NMDAR pore is blocked by an Mg2+ ion, the increase of EC Mg2+ has been used to block ionic flux. Similarly, the use of Cd2+ in the EC milieu has been employed to block Ca2+ flux, a widely used strategy to block Ca2+ flux through channels, as it mimics Ca2+ but is unable to flow. Some authors have substituted Ca2+ ions in the EC solution for Ba2+, which is able to flow through the NMDAR but does not activate IC signaling pathways and CICR. Additionally, the decrease of EC Ca2+ concentration or its depletion (Ca2+-free) in the EC media have been extensively used to evidence f-iNMDARs and/or, with the use of EDTA or 1,2-bis(2-aminophenoxy)ethane-N,N,N′,N′-tetraacetic acid (BAPTA), to chelate EC-free Ca2+. One strategy has been to clamp cell voltage to avoid depolarization and therefore avoid Mg2+ release from the NMDAR pore, thus blocking ion flux. Additionally, the IC loading of the Ca2+ chelator BAPTA or its membrane permeable version BAPTA-AM has been used to block IC Ca2+ actions given BAPTA fast chelating kinetics, thus avoiding CICR as a confounding variable. In addition, some molecular biology approaches have been used to reduce ion flux through the NMDAR. GluN1 subunit variants with point mutations of certain aa located in the conducting channel forming domain or in its vicinity have been generated. These point mutations include Asp 598, which has been changed to Gln (N598Q) or Arg (N598R), and Asp 616, which has been changed to Arg (N616R).

Finally, different strategies or inhibitors have been used to prevent the participation of Ca2+ IC pools in the response elicited by the NMDAR and therefore to dissect its flux-independent function. The ER depletion of Ca2+ by inhibition of the sarco endoplasmic calcium ATPase (SERCA) with ciclopiazonic acid (CPA) or thapsigargin (Thaps) has been employed, whereas the release of Ca2+ by CICR through the ryanodine receptors (RyR) has been inhibited with dantrolene or ryanodine (Ry). Similarly, the release of Ca2+ through the inositol tris-phosphate (IP3) receptors (IP3R) has been inhibited with xestospongin C (XesC). On the other hand, some authors used Ni2+ to exclude Ca2+ flux through T-type voltage-gated Ca2+ channels (VGCCs).

These approaches have been used by different groups to explore f-iNMDARs either in acute slices, organotypic cultures, cultured neurons, astrocytes, or heterologous cell systems. Since it is now evident that the NMDAR is able to elicit IC signaling independently of its channel function, the use of some of these strategies should become common for the study of the NMDAR.

## 3. Cellular Mechanisms Involved in f-iNMDARs and Its Cellular Effects

In the few dozen works that have reported f-iNMDARs, some pathways have already been demonstrated along certain cellular mechanisms. In addition, recent work has demonstrated conformational changes of NMDAR subunits in response to NMDA binding. In this core section, these aspects are reviewed. In Figure 2, the molecular mediators, pathways, and mechanisms that have been involved with GluN1 or GluN2 activation are represented. Like most new knowledge, f-iNMDARs has been controversial, also stimulated by apparent contradictory findings that may be related to subtle methodological differences [14,15]. Therefore, in this review, special attention is paid to some of these methodologies that in the opinion of the author could be relevant for f-iNMDARs cell biology, since some of those related to electrophysiology have been discussed [14]. 

### 3.1. Conformational Changes of NMDAR Subunits

Recently, in two papers published simultaneously by the group of Roberto Malinow, it was demonstrated that NMDA binding induced conformational changes of GluN1, independently of ion flux [16,17]. These authors used Förster resonance energy transfer-fluorescence lifetime imaging microscopy (FRET-FLIM) to measure the interaction between GluN1 subunits of the same NMDAR after NMDA treatment (25 µM) in cultured hippocampal neurons. Importantly, these authors experimentally ruled out that the FRET-FLIM observed came from subunits of different NMDARs, but instead from subunits of the same NMDAR [16]. The conformational change observed was not blocked by 7CK or MK-801, but it was blocked by APV, or an antibody (Ab) against the IC domain of GluN1 dialyzed into cells that presumably maintained the 3D conformation of these subunits. This conformational change, similar to electrophysiological responses of the NMDAR, was immediate, ligand-binding-dependent, and decreased in seconds after NMDA removal. A similar conformational change was observed when Glu was uncaged near a dendritic spine, again this effect was insensitive to 7CK but was blocked by APV. It is important to note that these results indicate that the conformational changes observed in GluN1 subunits, those tagged for FRET-FLIM studies, resulted in GluN2 subunits from ligand binding, implicating information transfer between GluN1 and GluN2 through conformational re-arrangements, as the author discussed. 

In a second paper [17], the same group demonstrated in cultured neurons that NMDA treatment reduced the interaction of GluN1 with Protein Phosphatase 1 (PP1), consistently with a conformational change of GluN1. In addition, this conformational change reduced its interaction with CaMKII, an effect that was dependent upon phosphatase activity and CaMKII dephosphorylation. Interestingly and in contrast with the conformational change of GluN1, the interaction of CaMKII with GluN1 did not return to baseline levels after ligand washout, suggesting that CaMKII could be further involved in molecular signaling after the NMDAR conformational change.

These results demonstrate that ligand binding to the GluN2 subunit is enough to induce a conformational change of the NMDAR. Intuitively, such conformational change is somehow required to initiate f-iNMDARs. Furthermore, the observation that the interaction of the NMDAR with molecular mediators is altered further supports f-iNMDARs.

### 3.2. Ca2+ Dynamics and LTD

One of the main areas of study in which f-iNMDARs has been involved is that of Ca2+ dynamics and its effects on LTD. In this sub-section, the evidence related to these cellular mechanisms is reviewed.

Some initial findings with cultured neurons described that NMDAR activation induced the release of Ca2+ from IC pools in response to NMDA (100 µM). In one of these works, the release of Ca2+ through RyR was evidenced with dantrolene, which reduced the iCa2+ increase by 33% [18]. However, the use of EC Cd2+ fully blocked the iCa2+ response, so it was assumed that the release of Ca2+ through RyR was produced by CICR, elicited by the entry of Ca2+ through the NMDAR. In a second work from a different group with hippocampal slices, dendritic Ca2+ released from IC pools was also evidenced with Ry or Thaps in response to tetanus stimulation at −35 mV [19]. This effect was sensitive to APV and was independent of VGCCs, which mediated somatic Ca2+ rise. The amount of Ca2+ released from IC pools through RyR constituted 65% of the total iCa2+ increase. With these observations, it was also assumed that CICR mediated the release of Ca2+ from IC stores, although the flux-independent release of Ca2+ from IC stores was not actually ruled out. 

Intriguingly, one group found in retinal Muller cells, a type of radial glia, that NMDAR activation increased phosphoinositide hydrolysis, eliciting a transient increase of IP3 [20]. However, this effect was strictly dependent on Ca2+ flux, since MK-801 or Ca2+ depletion in the EC solution depleted this effect. 

Two years later, while studying CaMKII role in the hippocampal synapse, it was observed in hippocampal slices that LTD elicited by low frequency stimulation (LFS) was not blocked by MK-801, suggesting its flux-independent nature [21]. The following year, another group using a double stimulation paradigm that induced Ca2+ store loading in slices and organotypic cultures found that Ca2+ release from the ER activated by afferent stimulation depended on such Ca2+ load [22]. This response was delayed relative to the MK-801 sensitive response, was dependent on IP3 and its diffusion, and was blocked by Thaps. Interestingly, the site of Ca2+ release from IC pools differed in their subcellular localization from that of Ca2+ entry through the NMDAR. However, these authors suggested that this response could be independent of the NMDAR and perhaps mediated by the metabotropic Glu receptor (mGluR), although they did not use APV nor mGluR inhibitors to rule this out. Nevertheless, these findings demonstrated that synaptically addressed Ca2+ stores exist but that they are insufficiently filled to provide appreciable Ca2+ increases or are inaccessible to triggering events at the synapse. In the same year, it was also found in slices that homosynaptic LTD was not blocked by MK-801 or IC Ca2+ chelation with BAPTA, whereas LTP was blocked. These authors found that LTD was dependent on PP1 and PP2 and independent from mGluR [23].

Afterwards, in hippocampal pyramidal cells in organotypic cultures with 0.1 Hz afferent stimulation, it was found that iCa2+ rise in spines is mainly dependent upon NMDAR activation that elicits Ca2+ exit from the ER, whereas EC Ca2+ played a minor role (10X less). VGCCs were not involved, as Ni2+ did not alter the response [24]. Three lines of evidence indicated that f-iNMDARs mediated iCa2+ rise in spines: the CPA abolition of Ca2+ transients, the Ry abolition of Ca2+ transients, and the circumvention of the Ry blockade by trains of stimulus that open NMDAR-mediated flux.

More recently, another group confirmed in acute hippocampal slices measuring field potential recordings of AMPA receptor that LTD induced by LFS depends upon Glu binding to the NMDAR, independently of Gly binding [25]. Glu binding elicited f-iNMDARs mediated by p38 MAP, involved previously in LTD. It was not blocked by MK-801 or 7CK, but it was blocked by APV. These results indicated that LTD does not require ion flux through synaptic NMDAR. In contrast, LTP elicited by high frequency stimulation (HFS) was blocked by MK-801 and APV. In addition, LTD was achieved in hyperpolarized cells, despite the Mg2+ block of the NMDAR. However, LTD required that basal Ca2+ levels are maintained, because lowering Ca2+ levels with BAPTA, applied in the IC solution of the electrode, potentiated AMPA-mediated synaptic responses. These authors suggested that a conformational change of GluN2 initiates IC signaling without a Ca2+ rise, countering the accepted view that LTD requires low levels of Ca2+ entry through the NMDAR. One year later, this group discussed the differences between their observations and those from other labs regarding the need of Ca2+ flux through the NMDAR for LTD [14]. Two years later, the same group demonstrated that such conformational changes do occur in the NMDAR [16] (see above).

Two years later, Kim and colleagues (2015) confirmed in hippocampal slices that LTD is induced despite chelating iCa2+ with BAPTA, but in contrast LTP depends upon Ca2+ entry through the NMDAR [26]. In the same year, it was found that LFS or low frequency Glu uncaging induced spine shrinkage in organotypic cultures [27]. This effect was blocked by the NMDAR Glu site antagonist CPP but not by 7CK. Similar to the work by Nabavi et al. (2013), p38 MAP was found to mediate this effect. These authors verified that 7CK inhibits postsynaptic currents and Ca2+ increase in response to Glu uncaging and confirmed that 7CK does not block LTD elicited by 15 min stimulation at 1 Hz in slices. Additionally, they found that high frequency Glu uncaging induced LTP, but in the presence of 7CK or MK-801 it induced spine shrinkage that was independent of mGluR. These authors explained that their finding of the non-ionotropic function of the NMDAR does not preclude the possibility that prolonged low Ca2+ rises induce LTD and spine shrinkage and suggests that, under physiological conditions, Ca2+ through NMDAR likely contributes to this kind of plasticity. Additionally, they propose that actin dynamics are involved in spine shrinkage and mention that PICK-1 and Arp 2/3 are required for LTD and spine shrinkage.

In addition to confirming the conformational changes of GluN1 described above, Aow et al. (2015) also verified that f-iNMDARs mediates LTD using 7CK to block NMDAR currents [17]. This is because in cultured neurons, NMDA (25 µM) induced a depression of frequency and amplitude of spontaneous synaptic events 15 min after NMDA washout. This effect was blocked by APV or by a dialyzed antibody (Ab) against the IC domain of GluN1 that presumably inhibits GluN1 conformational change. 

Additionally, with a protocol for induction of spike-timing-dependent LTD (t-LTD) in the somatosensory cortex, Carter et al. (2016) found that APV blocked it, whereas MK-801 did not. In addition, CPP blocked Ca2+ currents and t-LTD, whereas 7CK and 5,7DCK (a Gly-site-competitive inhibitor) blocked currents but not t-LTD [28]. Their findings show that non-ionic NMDAR signaling related to t-LTD is Glu-dependent. 

Recently, it was demonstrated in slices that LTP induced by theta burst stimulation (TBS-LTP) depends on APV and mGluR5, but not on mGluR1 [29]. In addition, these authors found that depotentiation achieved with LFS after TBS-LTP is proportional to LFS duration and that it is sensitive to APV and mGluR1 but insensitive to MK-801, Mg2+, and mGluR5. Therefore, depotentiation is mediated by f-iNMDARs. These authors explain that an equilibrium between phosphorylation and dephosphorylation of the NMDAR could provide the trigger for a conformational change in the NMDAR cytoplasmic domain, enabling its metabotropic function, and mention that p38 MAP could be involved, as it is central for depotentiation and for f-iNMDARs.

Last year, Abrahamsson et al. (2017) showed in slices that presynaptic NMDAR (preNMDAR) has specific IC signaling that regulates the spontaneous or evoked release of neurotransmitters [30]. They found that presynaptic frequency regulates excitatory post-synaptic potentials (EPSPs) through preNMDAR independently of Mg2+. With postsynaptic neuron at −80 mV and dialyzed with MK-801 (avoiding postNMDAR effect), APV reduced miniature EPSP (mEPSP) frequency, not amplitude. In addition, Ro 256981 (a GluR2B-competitive inhibitor) reduced mEPSP frequency, not amplitude and reduced spontaneous release, but MK-801 did not reduce mEPSP frequency nor amplitude, thus suggesting a flux-independent function of preNMDAR. Additionally, in a Rab interacting molecule-1 (RIM1) KO, they found that preNMDAR required RIM1 to regulate evoked release, but spontaneous release was independent of RIM. In this work it was also found that spontaneous release by preNMDAR was dependent on JNK2 function.

All together, these studies have shown that NMDAR is able to elicit flux-independent IC signaling that releases Ca2+ from IC pools that is independent of CICR. On the other hand, different groups have demonstrated that f-iNMDARs mediates LTD after LFS with p38 MAP as a critical mediator. Nevertheless, there seems to be no consensus yet on whether these observations neglect that low Ca2+ flow through NMDA would also be necessary for LTD, and whether preNMDARs are involved, and these issues are still matter of debate. Undoubtedly, the role of f-iNMDARs in LTD is a very relevant aspect given the importance of this mechanism of plasticity for CNS information handling and behaviour. The reader can refer to earlier reviews [6,7,8,9,14] in which the implications of this kind of NMDAR signaling have been analysed extensively.

### 3.3. Endocytosis, Signaling Pathways, and Membrane Dynamics

In a paper by Vissel et al. (2001), it was demonstrated that tyrosine dephosphorylation of the NMDAR occurs after repetitive application of Glu (1 mM) independently of ion flux in cultured hippocampal neurons and transfected HEK cells [31]. This tyrosine dephosphorylation of the NMDAR induced an amplitude decline of the electrophysiological response to an agonist, which was inhibited by the inclusion in the recording pipette of kinases Src, Fyn, or the phosphatase inhibitor dephostatin, without modifying single channel conductance or mean open time. The amplitude decline was dependent upon the number of applications, the time interval between them, and the agonist concentration, since 1 and 5 mM Glu generated this response, but 100 µM did not. This effect was mediated by AP-2 and dynamin and was given by a reduction of the number of surface NMDARs (66.9%), therefore indicating NMDAR endocytosis. This amplitude decline was observed even with IC BAPTA, with or without low EC Ca2+ (0.2 mM) and was not blocked by Mg2+, indicating its ion flux independency but was blocked by 7CK application. In addition, these authors identified the aa sequence and tyrosines in GluN1 and GluN2A IC domains that mediated the amplitude decline, and they determined that NMDAR internalization itself required Glu/Gly binding and not only Glu binding (given 7CK effect). In addition, they suggested that PTP1 and PLCγ could be involved in this effect after tyrosine phosphorylation. In light of these observations, the authors suggested that NMDAR has a memory mechanism linking its use and phosphorylation with its function that is independent of ion flux.

Meanwhile, other groups documented the activation of IC pathways by f-iNMDARs. In the work by Barria and Malinow (2002), it was found that agonist binding to the synaptic NMDAR is sufficient to bring the NMDAR containing GluN2A subunit (NMDAR2A) to the synapse and replace NMDAR2B, the mechanism related to synapse maturation [32]. Interestingly, APV and 7CK prevented the insertion of NMDAR2A but not of NMDAR2B into the synapse, whereas MK-801 did not have any effect, therefore also ruling out the participation of Na+ flux. These results confirmed that the synaptic NMDAR opening is not necessary for trafficking of NMDAR2A and supported a model in which ligand binding is required. 

Later, a different group found in slices that Gly binding to the NMDAR recruits the endocytic machinery preparing the NMDAR for clathrin-mediated endocytosis [33]. In this work, it was found that application of a high concentration of NMDA (1mM) + Gly (100 µM) elicited a depression of the electrophysiological response, an effect inhibited by the IC application of a recombinant SH3 domain of amphiphysin. In contrast, and similar to the observations by Vissel et al., a low concentration of NMDA/Gly (50 µM NMDA/1 µM Gly) did not generate this depression. Further experiments with Gly alone (100 µM) or with L689560 demonstrated that Gly was responsible for priming NMDAR endocytosis through the recruitment of the endocytic machinery, independently of ion-flux, increasing the association of the NMDAR with the AP2 complex, in particular with adaptin β2. Importantly, these effects were also achieved with the other Gly site agonist d-Ser with an even lower concentration (30 µM). However, as previously reported by Vissel et al., the internalization of the NMDAR itself required both NMDA and Gly binding after Gly priming and was not achieved by NMDA or Gly alone. With these observations, this group suggested that Gly could initiate transmembrane signal transduction independently of ionic flux.

In experiments with striatal cultured neurons, NMDAR signaling was achieved by ligand binding without flux; however, it required co-treatment of NMDA (1 µM) with DHPG (3 µM), a Glu metabotropic receptor agonist [34]. This co-treatment induced ERK1/2, Elk-1, and CREB phosphorylation as well as c-fos expression. This signaling was blocked by APV or MPEP, an mGluR5 antagonist, but not by MK801, a Ca2+-free EC solution, a BAPTA-AM, or an mGluR1 selective antagonist. In addition, NMDA/mGluR signaling was also independent of ER Ca2+ release and PLC/IP3, as Thaps and U73122, a PLCβ1 inhibitor, failed to block it, but it was dependent upon NMDAR-PSD95 interaction. ERK1/2 phosphorylation was also observed when EC Na+ was depleted, ruling out a role of Na+ flux through the NMDAR in this effect.

More recently, Li et al. (2016) reported in mouse hippocampal neurons that Gly induced the increase of AMPA mediated currents, miniature excitatory postsynaptic currents (mEPSCs), and fieldEPSP in a solution without Ca2+, with MK-801, EGTA, and Gly receptor inhibitor strychnine, a cocktail that fully blocked NMDAR ion flux [35]. This effect was mediated by ERK1/2 phosphorylation and the GluN2A subunit, but not by the GluN2B subunit. Additionally, it was found that the mutant form of GluN1 N598Q, which decreases Ca2+ flux, did not inhibit the Gly effect. Importantly, the threshold for the Gly effect was 25 µM, above the required concentration to activate NMDAR ionotropic function (10 nM; Johnson and Ascher 1987). Therefore, the authors proposed that this function may be relevant in certain pathological conditions when high Gly levels are achieved. These observations are consistent with the flux-dependent synaptic GluN2A-ERK link and the GluN2B extrasynaptic shut-off of ERK.

Last year, Ferreira et al. (2017) studied NMDAR dynamics in the cell membrane of cultured neurons and found that NMDAR2A diffusion and the area covered are slightly reduced with Gly, but it is conserved with enzymes breaking d-Ser and Gly [36]. In contrast, NMDAR2B diffusion and the area covered are significantly decreased with d-Ser, or the enzyme degrading Gly, but not with Gly. Additionally, the synaptic content of NMDAR2B was decreased with d-Ser, but not by Gly, whereas for NMDAR2A it was the same with both ligands. In addition, and consistently, d-Ser decreased NMDAR2B-PSD95 interaction. Additionally, d-Ser decreased FLIM between GluN1- subunits, but Gly did not. This effect was prevented by NMDA pre-treatment (20 µM). Together these findings support that Gly and d-Ser exert different signaling effects on NMDAR in a Ca2+ flux-independent manner.

Together these reports show that f-iNMDARs regulates NMDAR dynamics at the membrane, in particular its diffusion and endocytosis through different IC signaling mechanisms. Some of them elicited synergistically with mGluR, which results in different functional features. Notably, there is a close relationship between endocytosis and signaling. Intriguingly, some of these pathways and those involved in LTP are shared with ionotropic NMDAR signaling. 

### 3.4. Neuronal Death and Survival

A surprising area in which f-iNMDARs has been involved is that of neuronal death and survival. Traditionally, NMDAR Ca2+ flux-dependent signaling has been associated with both mechanisms, through the activation of pro-survival pathways activated by synaptic NMDAR activity or pro-apoptotic pathways and excitotoxicity elicited by extra-synaptic NMDAR. Though evidence has supported such segregation given by the NMDAR location, there are also some reports that have contradicted this idea. In principle, the location of the NMDAR in synaptic or extra-synaptic sites is related to the type of subunits assembled into the NMDAR: GluN2A for synaptic and GluN2B for extra-synaptic receptors and their IC pathways. In the following, the evidence that relates f-iNMDARs with neuronal fate is reviewed.

Early work shows that, in cortical cultured rat neurons, NMDAR pre-activation with Glu or NMDA (100 µM) potentiated cell death elicited by Glu (30-50 µM), presumably independently of Ca2+ influx, as Ca2+-free media was used in these experiments [37]. However, the researchers did not rule out (nor suggest) that Na+ flux could be involved. They advocated Ca2+ flux-independent, NMDAR-dependent, mitochondrial Ca2+ uptake and suggested that Ca2+ rise location is important for final response. Their results supported that NMDAR has two different categories of pathways: a Ca2+-dependent pathway and a Ca2+-independent pathway that facilitates Ca2+-dependent cell death. These observations are consistent with the recent work by Weilinger et al. (2016 see below). Later, a group found that blocking NMDAR ionotropic function with a high concentration of MK-801 failed to block ischemic changes in dendritic structure or rapid elevations of iCa2+-induced in a mouse global ischemia model. In contrast, Ca2+ flux through the NMDAR was blocked, thus suggesting that f-iNMDARs could be involved [38].

More recently, it was found that Gly treatment of cultured cortical neurons induced phospho-Akt (pAkt) in Ca2+-free media with EGTA, MK-801 pretreatment, or EC BAPTA [39]. This effect was independent of strychnine-sensitive Gly receptors and p38 MAP, implicated previously in NMDAR-dependent LTD. The same effect was observed in HEK cells transfected with GluN1/GluN2A subunits, but not with GluN1/GluN2B subunits, nor with GluN1, GluN2A, or GluN2B alone, although GluN1 traffic to the cell membrane was not granted. pAkt still increased with the mutant forms of GluN1, GluN1N598Q and GluN1N598R, or with an shRNA against GluN2B, but decreased with an shRNA against GluN2A. In addition, the Gly competitive antagonist L689560 decreased pAkt, whereas d-Ser also induced pAkt. Interestingly, Gly prevented neuronal death after Glu (100 µM)/Gly (1 µM) treatment, beyond the protective effect of MK-801, an effect reverted by L689560. These authors discussed that previous reports found that f-iNMDARs was insensitive to Gly, contrary to their findings, and proposed that GluN1 induces a conformational change of GluN2A but not of GluN2B. However, it is possible that this is related to the fact that they measured pAkt, which is activated by synaptic (NMDAR2A) signaling. These authors did not explain where could Gly comes from in a physiological setting but argued that neuroglial cells may be the source in ischemia and pointed out that d-Ser also induced pAkt. Notably, it has been demonstrated that d-Ser of astroglial origin regulates synaptic plasticity and modulates LTP [40,41].

Consistent with these results, the same group reported in a second paper that Gly protects rats against middle cerebral artery occlusion (MCAO) in the presence of MK-801/strychnine, reducing the infarct volume and neuronal death [42]. Importantly, this effect was achieved even when Gly was injected up to 6 h after MCAO. The Gly site inhibitor L869560 reduced this Gly neuroprotective effect that was also sensitive to the Akt inhibitor IV. Correspondingly, the use of both inhibitors decreased the amount of pAkt. Moreover, Gly improved the performance of animals in neurobehavioral tests, an effect blocked by L869560 or inhibitor IV. With these findings, these authors proposed that Gly could be used as therapy for stroke.

A different group recently demonstrated that binding of NMDA to NMDAR also elicits flux-independent actions involved in excitotoxicity [43]. This group found in hippocampal neurons of acute slices that NMDA (100 µM) alone caused cell blebbing that was blocked by APV and CGP-78608, an inhibitor of the NMDAR Gly binding site, but was insensitive to MK-801. They also found that NMDA elicited a secondary Ca2+ influx (Ca2+ 2i), which is insensitive to MK-801 and BAPTA applied through the electrode, that was blocked by APV/CGP and by an EC blocking peptide of Panx1 (10panx). Since the Ca2+ 2i mediated by Panx1 required Glu and Gly binding, the authors suggested that Gly or d-Ser is present in their preparation. In addition, a pre-block protocol that eliminated NMDA puff responses with MK-801 did not block the Ca2+ 2i, although NMDA elicited currents, presumably because of the slow open pore block of MK-801. Additionally, GluN1 N616R transfection did not block Ca2+ 2i, and MK-801+BAPTA blocked blebs, but not Ca2+ 2i, so flux through NMDAR is required for blebbing but not for the Ca2+ 2i. Additionally, the Ca2+ 2i was observed at −60 and −80 mV, with Mg2+, TTX, and CNQX supporting its flux-independent nature and its independency from neuronal AMPAR activity. Furthermore, GluN1 co-immunoprecipitated Panx1 and Src under basal conditions, whereas NMDA/Gly treatment increased Src–NMDAR interaction and tyrosine 308 phosphorylation (pY308) of Panx1. In addition, the peptide Src48 blocked the Ca2+ 2i and blebbing. Additionally, this group found that pY308 of Panx1 depended on Src–NMDAR interaction, APV, and CGP, but not on BAPTA or MK-801, and that it induced Ca2+ 2i and blebbing. In addition, under oxygen-glucose deprivation (OGD), they found that iCa2+ increase was blocked by MK-801 (58%), by APV (92%), and by TAT–Panx308 (which blocks Src–Panx1 interaction) (78%). Additionally, MK-801 did not alter neuronal death, but TAT–Panx308 did and with MK-801 reduced lysis to APV levels. Thus, it was concluded that NMDAR and Panx1 mediated Ca2+ entry, but Panx1 is critically linked to neuronal death. Additionally, OGD decreased mitochondrial membrane potential (mΔψ), related to mitochondrial permeability transition pore (mPTP), which was prevented by TAT–Panx308 or 10panx. TAT–Panx308 ameliorated in pre- or post-treament MCAO damage, improving behavioral performance. After two weeks, TAT–Panx308 pre-MCAO treated rats recovered sensorimotor function. With these observations, these authors suggested that these unexpected f-iNMDARs may play a role in the failure observed with NMDAR inhibitors that have been used to treat excitotoxicity in some pathological conditions.

Taken together, these findings demonstrated that ligand binding to either the Glu or the Gly sites elicits specific flux-independent signaling, even with opposing actions in terms of neuronal survival. As expected, the authors of these works suggested using this novel knowledge for the design of new therapies oriented to treat pathologies in which neuronal death is a critical feature. Interestingly, the findings by Weilinger et al. (2017) show that f-iNMDARs activates other channels that mediate Ca2+ entry, thus raising the possibility that cell membrane molecular clusters represent an additional level of complexity for f-iNMDARs. 

### 3.5. Pathologies

One group found in hippocampal neurons of slices that synaptic depression induced in minutes by oligomeric Aβ (oAβ) is not relative to presynaptic elements, but instead that it is dependent upon synaptic stimulation and activation of the NMDAR [44]. They found that APV blocked the oAβ-mediated synaptic depression at −60 mV, therefore suggesting that it was a f-iNMDARs. Ifenprodil blocked the effect of oAβ, thus suggesting that NMDAR2B mediate synaptic depression. The oAβ effect was not mediated by the release of the Mg2+ block or by modifying the NMDAR opening. Consistently, though MK-801 blocked the synaptic activity at +40 mV, it did not block oAβ synaptic depression. These results demonstrated that oAβ synaptic depression involves f-iNMDARs.

Almost simultaneously and independently, Kessels et al. (2013) consistently found that NMDAR2B but not NMDAR2A is required for oAβ-induced synaptic depression [45]. Additionally, they found that oAβ leads to a selective loss of synaptic NMDAR2B responses and promoted the switch of subunit composition from NMDAR2B to NMDAR2A. The effect of oAβ was blocked by APV but not by MK-801 or 7CK and required NMDAR2B activation, as Tamburri et al. (2013) also observed. Interestingly, a tyrosine phosphatase inhibitor blocked oAβ-induced synaptic depression. These data, again and in agreement with the independent work by Tamburri et al., support that oAβ synaptic depression is mediated by f-iNMDARs.

Given the relevance of the NMDAR in CNS function in physiological and pathological conditions, the findings described here that involve f-iNMDARs with AD open the possibility that this signaling is involved in other pathologies related to NMDAR. 

## 4. f-iNMDARs in Astrocytes

As expected, most of the work made with f-iNMDARs has been performed in the synaptic/neuronal context. This is because this receptor plays a fundamental role in synaptic communication, and the neurocentric theory assumed for decades that information in the CNS is handled only by neuronal cells. Nevertheless, this receptor is also expressed in other cells of the CNS and indeed in cells and tissues beyond the CNS. In the last three decades, the expression of the NMDAR in astroglia has been a matter of intense research and debate [3]. This debate has been mainly fuelled by the fact that in neurons the NMDAR is assumed to allow ionic flux only after membrane depolarization, which in turn removes the Mg2+ block. In contrast, it has been assumed for a long time that astrocytes are not “electrically excitable” cells. Thus, this notion posed a conceptual barrier to accepting that NMDAR could be functional in these cells [46]. However, different groups in the last 15 years have provided strong evidence that tissue astrocytes do possess functional NMDARs. It turns out that astroglia have NMDARs assembled with different GluN2 subunits that provide particular biophysical features to these receptors, including insensitivity to Mg2+ with the participation of GluN3 subunits [47]. 

The debate on the existence of NMDARs in astroglia was also fuelled by initial works with cultured astrocytes in which there was no electrophysiological response to NMDA [48]. These findings lead to a consensus by the end of the 1990s, indicating that these receptors were not expressed by cultured astrocytes [49]. However, by the end of the 1990s, a group published results in which membrane currents and an increase in iCa2+ were observed in human white matter astrocytes in response to NMDA (1 mM) [50]. Oddly, and against the common pharmacology of this receptor, APV did not block these responses, and in Ca2+-free solution the iCa2+ response was reduced by only 26%. In addition, Thaps and CPA augmented the currents, suggesting that Ca2+ from IC pools was somehow involved along with a crosstalk between them and the cell membrane. Moreover, currents and iCa2+ increase were both sensitive to guanosin 5′-[β-thio]diphosphate (GDPβ-S), a G-protein inhibitor. In a second work from the same group, it was suggested that the NMDAR could be regulated by G proteins because GDPβs inhibited 82% of currents in response to NMDA [51]. The possibility that GDPβs could inhibit signaling directly elicited by NMDARs was not ruled out, and it could be the first hint that, in astrocytes, the NMDAR could activate f-iNMDARs. 

More recently, we and others published results indicating that in cortical rat cultured astrocytes the NMDAR may work in a flux-independent manner. Gerard and Hansson published a paper in which they described that rat cortical astrocytes co-cultured (9–11 days) with endothelial cells presented iCa2+ responses to NMDA (100 nM–100 µM) [52]. These responses were fully blocked by APV or ifenprodil, indicating that the NMDAR with the GluN2B subunit, a subunit that was detected by immunofluorescence, mediated these responses. Interestingly, the amplitude of these responses was blocked only partially (50%) in a Ca2+-free solution or with Cd2+. Moreover, Ca2+ depletion from IC stores with caffeine and Thaps almost fully blocked (90%) response amplitude, whereas the combination of Ca2+ depletion from IC pools and EC Ca2+-free solution fully blocked the response. These results indicated that the source of Ca2+ were mainly the IC pools. Consistently, XesC diminished iCa2+ response amplitude significantly (80%) and, in combination with Ca2+-free EC solution, reached more than 90% inhibition. These results constituted the first suggestion that in cultured cells f-iNMDARs occurs beyond its ionotropic function, although these authors mentioned that their results did not rule out the possibility that CICR was involved.

Later, an iCa2+ rise elicited by NMDA (1–100 μM) in rat cultured cortical astrocytes was observed [53]. Curiously, the NMDA effect was delayed by 500–1000 s after its application, presumably due to the low permeability of the NMDAR in astrocytes, or to its metabotropic-like mechanism. The NMDA effect was partially sensitive to the EC Ca2+-free solution, suggesting also an NMDAR canonical ionotropic function, but it was fully blocked in Ca2+-free solution in combination with Thaps, evidencing also the Ca2+ release from IC pools, although CICR was not ruled out. Moreover, an iCa2+ rise was sensitive to U73122, an inhibitor of phospholipase C (PLC), suggesting that IP3 synthesis could be involved. Interestingly, these authors also found long-term NMDA (20 μM, 8 h) effects blocked by MK-801. They observed the activation of the PLC/protein kinase C (PKC)/p35/cyclin-dependent kinase (Cdk5) pathway that leads to nuclear factor erythroid 2-related factor (Nrf2) activation, its nuclear accumulation, and the transcription of its target genes. 

The same year, we serendipitously found f-iNMDARs in rat cultured cortical astrocytes [15]. In these cells, the expression of the seven NMDAR subunits was found at the mRNA and protein level by immunofluorescence and of GluN1 by WB. Interestingly, we found that acid–NMDA (1 mM; pH 6.0) elicited iCa2+ responses in astrocytes that were not blocked by MK-801 nor by Ca2+-free EC solution, but they were blocked by APV, KYNA, XesC, Ryanodine, or GluN1 knockdown by siRNA. These results suggested that CICR did not mediate iCa2+ rise because a) IP3R inhibitor almost fully blocked the response, suggesting that IP3 is involved, and the second messenger is not related to CICR, and b) the Ca2+ rise occurred in Ca2+free media or in the presence of MK-801. Our findings are consistent with those by Kettenman and Schachner (1985), who did not find electrophysiological response in these cells, but contrast with the co-culture model employed by Gerard and Hansson, who observed some iCa2+ rise in Ca2+-free media. Our unpublished results have shown that iCa2+ response was elicited by acid pH, which indeed inhibits NMDAR canonical function [1,11], rather than NMDA itself. In addition, acid–NMDA treatment depleted mitochondrial membrane potential (mΔψ). This effect was strictly dependent on NMDA and acid pH as it was blocked by MK-801 and not elicited by acid pH alone. These observations suggest that there is some sort of functional segregation between Ca2+ flux-dependent and -independent functions of the NMDAR in cultured astrocytes, which is supported by our IF and WB experiments, similar to the findings of Kato and Murota (2005).

Taken together, these findings support the notion that, in cultured astrocytes, f-iNMDARs and flux-dependent signaling coexist. However, whether tissue astrocytes have f-iNMDARs beyond the ionotropic function is still unknown. Nevertheless, in cultured human astrocytes, the canonical function of the NMDAR has been reported [54]. For a discussion on the implications for these findings, the reader can refer to our recent open access review [3].

## 5. Insights

The organization of neuronal transmission into ionotropic and metabotropic transmission was established by the end of the 1970s [55]. Ionotropic transmission was clearly differentiated by its short latency and the increase of conductance due to the opening of “ionic gates” in the postsynaptic cell, along with its short change in membrane potential. On the other hand, metabotropic transmission was barely known, but it was assumed to indirectly trigger chemical reactions in the postsynaptic cell without affecting the “ionic gates” and therefore keeping conductance intact. Since then, this paradigm has dominated the thinking and development of neuroscience. Nevertheless, a couple of decades ago, it was found that ionic flow through ionotropic receptors also elicits metabotropic-like signaling, activating chemical reactions in the postsynaptic cell. Moreover, today there is a good amount of evidence demonstrating that ionotropic receptors are also able to activate IC signaling pathways independently of ion flux. This has been found not only for NMDAR but also for kainate and AMPA Glu receptors, nicotinic acetylcholine receptors [56], VGCCs [57], and potassium channels [58]. The new knowledge suggests that the partition between ionotropic and metabotropic transmission has some inconsistencies, as they share features not previously acknowledged. With these considerations, in the following, insights are presented regarding structure–function, signaling, pathology, and evolution.

### 5.1. Structure–Function

With the framework of f-iNMDARs it is interesting to re-examine NMDAR from the cell biology perspective to further gain insight of its structure–function relationship. In this regard, the size of the NMDAR, when compared with peptides able to form voltage-sensitive transmembrane channels such as gramicidin, is noteworthy. This peptide has 15 aa with a hydrophobic nature that forms a helical structure in lipid bilayers and after dimerization it forms a cation selective ion channel with measurable conductance and lifetime [59]. This size comparison brings into consideration how complex the unknown functions may be within the 1000–1400 aa that form NMDAR subunits, beyond those already described for ligand binding, regulation, and EC and IC molecular interactions, among others.

A notable and interesting finding made by different groups is that specific f-iNMDARs is elicited depending on which subunit is activated. Some works have reported distinct f-iNMDARs associated with the binding of the agonist or the co-agonist, or mediated by a specific subunit. On the other hand, some authors have identified flux-independent effects that may or may not be blocked by competitive inhibitors of GluN1, but are systematically inhibited by GluN2-competitive inhibitors. These observations suggest that f-iNMDARs can be activated by an agonist or co-agonist, but that they require information transfer among subunits to allow the NMDAR ensemble to elicit IC signaling, as discussed by Dore et al. (2015). This phenomenon was suggested by different groups and has recently been demonstrated, because the activation of f-iNMDARs with NMDA alone induces the conformational change of GluN1. Moreover, well-known mechanisms such as Gly desensitization of the NMDAR and Gly or d-Ser potency depending on GluN2 identity indicate that information transfer occurs among GluN subunits [11]. Interestingly, such information transfer may occur bidirectionally, from GluN2 to GluN1 or from GluN1 to GluN2A, although it has not been reported to GluN2B. However, it is possible that this apparent lack of information transfer from GluN1 to GluN2B could be due to the biological effect analyzed. On the other hand, the blockage of f-iNMDARs with Gly inhibitors, beyond APV, in response to GluN2 agonist and the absence of exogenous co-agonist, has led the authors to suggest that the co-agonist could be present endogenously in their preparations. However, though it has been suggested that the active co-agonist could be related to the localization of the NMDAR, it has also been proposed that saturating concentrations of co-agonist are unlikely [11]. Nonetheless, it is possible that these observations could alternatively imply that f-iNMDARs by the ensemble are not only elicited by the agonist or co-agonist, but that it may also be blocked by competitive inhibitors of the agonist or co-agonist, without actually competing. In other words, the competitive inhibitor is able to avoid the information transfer among subunits perhaps through inhibiting conformational changes of the secondary subunit (locking it into a non-active state) required for ensemble f-iNMDARs. This effect of competitive inhibitors would be additional to the avoidance of conformational closure of the LBD of the bound subunit that is known to inhibit channel opening [11]. However, more work is needed to test this notion, implicating that inhibitory information is also transferred from one subunit to the other.

### 5.2. Signaling

Undoubtedly, the role of f-iNMDARs in LTD is a very relevant aspect given the importance of this mechanism of plasticity for CNS information handling and behaviour. The reader can refer to earlier reviews [6,7,8,9,14] in which the implications of f-iNMDARs has been analyzed extensively in the context of LTD.

Perhaps the most astonishing side of f-iNMDARs in terms of cell biology is that several IC mediators involved in it have been shown to mediate flux-dependent signaling. That is the case of Akt, CREB, ERK1/2, Elk-1, c-fos, and p38 MAP, activated indirectly in response to Ca2+ flow through the NMDAR, which in turn activates calmodulin or calpain. This phenomenon raises the thrilling possibility that the NMDAR function, which has been associated with its ionotropic function, instead also comprises flux-independent actions that have been underestimated for decades. 

Also surprising is the fact that f-iNMDARs can discriminate between co-agonists Gly and Ser, which are able to modulate NMDAR membrane dynamics differently. In contrast, NMDAR channel function seems to be activated by both co-agonists with similar potencies, although the GluN2 subunit assembled into the NMDAR influences it, with NMDAR2D being the most sensitive to these co-agonists, one of the subunits suggested to conform astrocyte NMDAR [47]. Nonetheless, differences exist between Gly and d-Ser actions at the synaptic level, but they seem to stem from ligand availability rather than differential signaling [60].

Other engaging facet of f-iNMDARs is that different groups have found that it may act synergistically with mGluR. In in vivo conditions NMDAR and mGluR activation may occur concomitantly after Glu vesicular secretion. In this regard, it has also been shown that NMDAR interacts with dopamine receptors, physically and at the signaling level [61]. Therefore, it is possible that f-iNMDARs could be in some cases an emergent property resulting from interactions with G protein coupled receptors; however, more evidence is required to reach these conclusions.

An intriguing aspect regarding f-iNMDARs is the wide range of Glu or NMDA concentrations that have been employed to unmask f-iNMDARs. This range encompasses from µM to mM concentrations. Typically, low µM concentrations (10-50) are employed to elicit NMDAR flux in different models with neurons, with concentrations in the hundred µM (100–500) to mM employed to induce excitotoxicity, although the temporality is also relevant. Nevertheless, it has been demonstrated that Glu concentration in the synaptic cleft may reach the mM range (1 mM) for several ms, but of course this is dependent on the level of synaptic activity, whereas in pathological conditions constant higher levels can be reached [62]. Moreover, simultaneously with Glu vesicular secretion at the synaptic cleft, there is also an acidification that reaches pH 6–6.4 near the presynaptic membrane, therefore making feasible some acidification near the postsynaptic membrane [63]. This acidification results because synaptic vesicles have low pH, since vacuolar H+ ATPase generates an H+ gradient that is profited by vesicular Glu transporter to translocate Glu into the vesicle. Moreover, acidic pH is well known to inhibit NMDAR ionotropic function, but conversely it seems to elicit f-iNMDARs, as we observed in cultured astrocytes. Therefore, more research is needed to further elucidate how agonist concentration and synaptic cleft acidification govern NMDAR flux-dependent and -independent functions at the synapse, extra-synaptic sites, and glial cells in physiological and pathological conditions.

In addition, if the wide expression and function of the NMDAR in non-neuronal cells and other tissues beyond the nervous system is taken into account, the f-iNMDARs also calls for a reconsideration of how we understand the biology of this receptor. This is because it is possible that in some cells f-iNMDARs could be more relevant than recognized. To illustrate this idea, consider the immune cells in which NMDAR has been detected. In contrast with cerebro spinal fluid (CSF), Glu levels in blood are 40× higher (40 µM vs. 1 µM) [64]. Thus, it seems likely that the NMDAR has other regulatory and functional mechanisms, since its steady channel activation would lead to iCa2+ levels that could alter cell homeostasis or generate cell death. Also supporting this notion, it has been known for decades that cultured astrocytes are resistant to neurotoxic Glu levels or that acidification does not decrease their ATP levels [64,65].

Despite the wide diversity of the NMDAR that could potentially be assembled with the repertoire of GluN2 and GluN3 subunits, most of the work made with this receptor has focused on NMDAR2A and NMDAR2B. Nevertheless, within the CNS, astrocytes that have turned into relevant players of information handling and synaptic plasticity express GluN2D, GluN3, and high amounts of GluN2C [3]. Nonetheless, despite their diversity, the functional properties of the NMDAR in these cells are studied in very few types of astrocytes. In white matter and grey matter human astrocytes, atypical observations have been made. Furthermore, NMDAR function in these cells was precluded by the a priori expectations that NMDAR should work as their neuronal counterpart [46]. Thus, though there are today some apparent inconsistencies observed in f-iNMDARs, it could be inaccurate to disregard this kind of signaling, mainly considering the complexity of this receptor, particularly if it is considered that different approaches and independent groups have substantiated the idea that an old known synaptic plasticity mechanism as LTD is mediated by f-iNMDARs.

### 5.3. Pathology and Clinic

Another aspect in which f-iNMDARs is involved is that of human pathologies. In particular, two independent groups have gathered evidence demonstrating that oAβ elicits f-iNMDARs that leads to synaptic depression. The question that follows is whether other pathologies could be mediated by f-iNMDARs. In this regard, our unpublished observations data suggest that this kind of signaling could also be involved in the neuronal effects associated with encephalitis by antibodies against NMDARs (in preparation).

The f-iNMDARs also creates the possibility of further analyzing the known clinical failure of NMDAR inhibitors to prevent neuronal death in pathological conditions, as discussed by Rong et al. (2016) and Weilinger et al. (2016) The incomplete knowledge of the NMDAR structure–function must be involved, including differences and specificities of flux-dependent and -independent signaling. In this regard, the increase of iCa2+ mediated by pannexins leading to cell death or the activation of Akt by Gly that prevents it evidence this incompleteness. In addition, the flux-dependent and -independent signaling of the NMDAR must be involved in the non-equivalent effects of NMDAR inhibitors for preventing cell death [66,67], beyond their mechanism of inhibition. On the other hand, it is also possible that excitotoxicity mediated by the NMDAR is a common secondary effect for, instead of a main cause of, different pathologies in which neuronal death occurs. This could also help to explain the clinical failure of NMDAR inhibitors to treat neurodegenerative diseases. The work of Weilinger et al. (2016) also brings to attention the role of supramolecular clusters and the relationships among the molecules comprising them, helping us further understand NMDAR cell biology, for which the interacting proteome should be critical [68,69], although it may vary among different NMDAR ensembles or cell types.

### 5.4. Evolution

From the evolutionary angle, the high protein sequence conservation (circa ≈95% of aa identity) between human and rat GluN1 subunits is noticeable. This identity contrasts with other membrane receptors less conserved between the same species. For instance, IL-2Rβ presents ≈35% aa identity. This aa sequence conservation of GluN1 among these mammal species suggests that the primary structure of this subunit plays a critical role for its structure–function. On the other hand, phylogenetic analyses have suggested that an ancestor of ionotropic Glu receptors was evolved from a primitive signaling mechanism that already existed prior to multicellularity, before the existence of plants and animals, but that such ancestor did not work primarily as a channel. Rather, these analyses have suggested that the channel activity evolved after, with the first actual NMDAR with a postsynaptic function emerging in the cnidarian group [70,71]. Interestingly, and supporting this notion, the GluD2 member of the ionotropic Glu receptors, which is similar to AMPA and kainate receptors, do not function as an ion channel but is involved in LTD and synapse formation at the parallel fiber-Purkinje interaction in the cerebellum, the ancient brain [72]. Further phylogenetic analysis of ionotropic Glu receptors including these members could shed light on the structure–function relationship of these molecules and their evolution. Taken together, these evolutionary traits of the NMDAR and ionotropic Glu receptors, along with the flux-independent function present in other channel families, strongly suggest that f-iNMDARs is ancestral, present in other non-mammal taxa with related ionotropic-like Glu receptors, and conserved through evolution, further supporting that f-iNMDARs may have a more relevant role than acknowledged. Interestingly, the dual signaling function of the NMDAR is also consistent with the natural selection of multi-functional molecules, known to occur with respect to cell membrane molecules [73,74]. Ultimately, it would be interesting to analyze the evolutionary conditions that favoured the emergence of the NMDAR, that requires agonist and co-agonist binding for its opening, a feature unique among Glu receptors, and to assess its advantage for the fitness of cnidarians and the groups that conserved it.

## 6. Conclusions

Mounting evidence obtained by many different independent groups with distinct approaches and models reviewed here have demonstrated that the NMDAR is able to elicit flux-independent signaling separately from the well-known ionotropic function. An understanding of f-iNMDARs increases the complexity of the NMDAR biology and recalls for a reconsideration of how we understand this channel. Given the ubiquitous expression of the NMDAR in a diversity of cells within and beyond the CNS, it is possible that this function has a more prominent role than expected. More importantly, f-iNMDARs has been demonstrated to mediate different cellular mechanisms such as LTD, Ca2+ release from IC pools, endocytosis, cell membrane molecular dynamics, pH sensing, cell death and survival, or neuronal function in pathological conditions. Although there are still some inconsistencies regarding the IC pathways activated, it is clear and astonishing that some IC molecules, mainly kinases, known to mediate canonical flux-dependent NMDAR signaling, also mediate f-iNMDARs. Nevertheless, more research is needed to further dissipate these inconsistencies and other questions regarding its precise role. Be it as it is, the f-iNMDARs raises the thrilling possibility that the NMDAR function, always studied in the context of its neuronal ionotropic activity, also comprises flux-independent actions that have been underestimated for decades.

An emerging feature of f-iNMDARs is that apparently it can either be elicited by one type of subunit acting alone or that it requires NMDAR ensemble activity precising information transfer between subunits. Although the action of a single subunit could actually be the result of the constraints imposed by the experimental outcome measured, the ensemble activity could have as a consequence a signaling blockade by competitive inhibitors without actually competing with the ligand, but instead locking the subunit and avoiding its conformational change, the information transfer, and in turn the signaling. However, more experiments are required to test this idea. 

Finally, the ubiquitous expression of the NMDAR in cells and tissues beyond the CNS, residing in EC milieus chemically different to the CSF, together with some evolutionary traits of this receptor, support the notion that f-iNMDARs may have a more relevant role than that acknowledged today, although more work is needed to further test this idea.

## Figures and Tables

**Figure 1 ijms-19-03800-f001:**
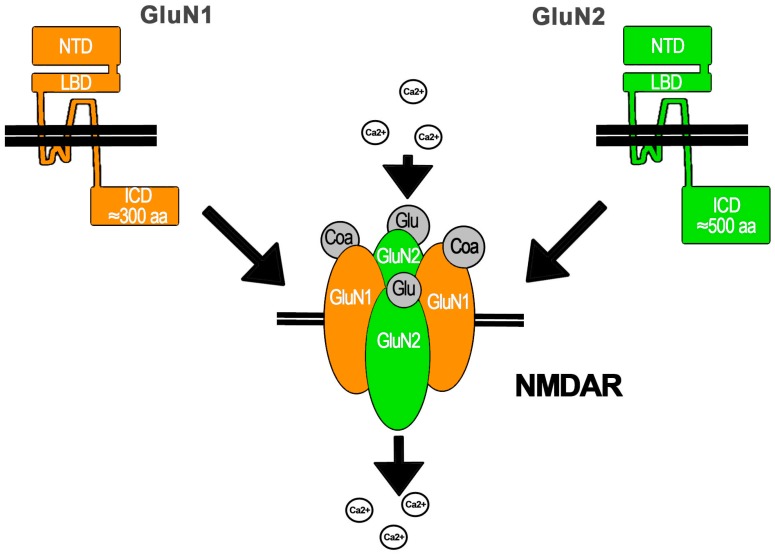
General structure of N-methyl-d-aspartate (NMDA) receptor (NMDAR). The NMDAR is assembled with GluN subunits, of which two GluN1 units are obligate. NMDAR subunits have the same general structure with NTD, LBD, IC domain, and three transmembrane helices with two loops, the IC that re-enters into de cell membrane. Glu binds GluN2 subunits, whereas Gly or d-Ser bind GluN1 (or GluN3; not shown) subunits. After agonist and co-agonist binding and Mg2+ release by membrane depolarization, the channel opens and enables Ca2+ flow (for details, see text).

**Figure 2 ijms-19-03800-f002:**
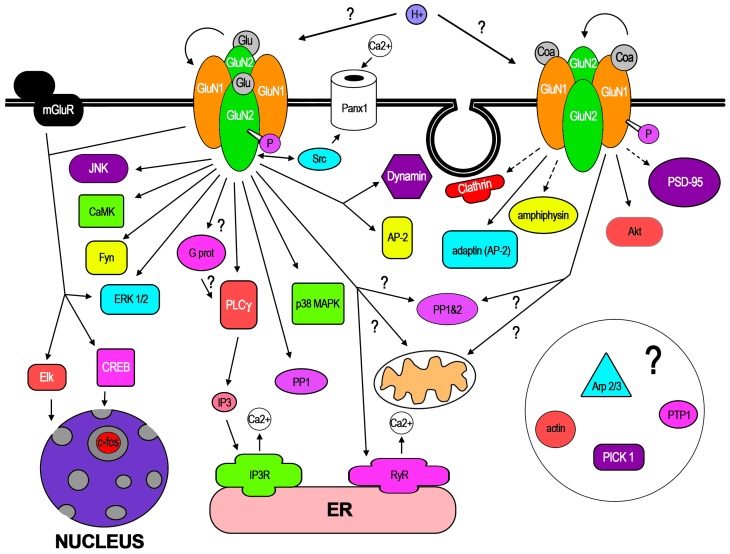
Molecular mediators, pathways, and organelles involved in f-iNMDARs. The molecular mediators pathways and organelles associated with f-iNMDARs are organized into those activated by the GluN2 subunit (*left*), those activated by the GluN1 subunit (*right*), or those not yet determined (*center; arrows with question marks*). In the case of G proteins activated by the GluN1 subunit, question marks indicate that only indirect evidence has been gathered. For extracellular H+, the question marks indicate that it is not clear which subunit mediates the induction of f-iNMDARs. Bracketed lines indicate an indirect relationship, such as the relationship with PSD-95 (known to bind GluN2 subunits) that is induced by GluN1 activation. The arrow loops from one subunit to the other indicates transfer of structural information between subunits. Those mediators only inferred to participate are encircled in the bottom right corner with a question mark (for details, please refer to text).

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
