# Peer review of "Flux-Independent NMDAR Signaling: Molecular Mediators, Cellular Functions, and Complexities"

_ijms, 2018, doi:10.3390/ijms19123800_

Round 1

Reviewer 1 Report

This review by Montes de Oca summarizes and puts in context the current information about the "flux-independent" function of NMDA Receptors (NMDARs). Although evidences for this "non-ionotropic" role for NMDARs were reported decades ago, it has been in the last 4-5 years when it has been confirmed and become accepted by the scientific community. This review, therefore, is timely and it will be of high interest in the field.

The author did a great job in selecting and explaining the published work in which this flux-independent NMDAR function has been probed or tested. I have, therefore, just a few minor comments:

1) Section 4: Approaches to study f-iNMDAR: I think this is a very valuable section and it really helps the reader to understand the work reviewed by the author. Therefore, in my opinion, it should appear much earlier in the review. I worry that the non-expert reader may become a bit lost with the details provided in the first part of the review, and having that information in advance, would help them throughout the reading.

2) I'm wondering if the author can comment or, at least, mention other ionotropic receptors with "non canonical" (or metabotropic) function, such as kainite receptors. That would provide the reader with a better background to understand that this flux-independent NMDAR function is not unique in the brain.

3) I would suggest the author include some references (if the space available permit it), especially in "NMDAR essentials" section. A recent review about NMDAR structure and pharmacology by Traynelis (PMID: 33037851) , should probably be cited. In addition, some refs are duplicated in the reference list. (eg, refs 2 and 10)

4) Page 5 . Line 8: There is a typo in the NMDA concentration: it should be 25 microM, an not mM

Author Response

1)Section 4: Approaches to study f-iNMDAR: I think this is a very valuable section and it really helps the reader to understand the work reviewed by the author. Therefore, in my opinion, it should appear much earlier in the review. I worry that the non-expert reader may become a bit lost with the details provided in the first part of the review, and having that information in advance, would help them throughout the reading.

This change has been done.

2) I'm wondering if the author can comment or, at least, mention other ionotropic receptors with "non canonical" (or metabotropic) function, such as kainite receptors. That would provide the reader with a better background to understand that this flux-independent NMDAR function is not unique in the brain.

Other ionotropic receptors that are known to elicit flux-independent functions are now mentioned in page 14 lines 29-31.

3) I would suggest the author include some references (if the space available permit it), especially in "NMDAR essentials" section. A recent review about NMDAR structure and pharmacology by Traynelis (PMID: 33037851) , should probably be cited.

Thank you very much for this reference, I was not aware of it and of some of the recent conceptions of NMDAR structure-function. This reference is now cited in the “NMDAR essentials” and “Insights” sections, given its relevance.

In addition, some refs are duplicated in the reference list. (eg, refs 2 and 10).

This has been corrected.

4) Page 5 . Line 8: There is a typo in the NMDA concentration: it should be 25 microM, an not Mm

This has been corrected.

Reviewer 2 Report

This review summarizes the current knowledge on flux-independent NMDARs signaling, taking into consideration neuronal and non-neuronal cells. Although the topic is of great interest, reading this review is difficult in its current state. I believe that a moderate revision will improve the manuscript.

Major issues:

1) English should be revised.

2) The main concern is that the review is mostly a summary of different papers, often merely reporting results in details, and adding the authors' conclusions. In this way, the review is difficult to read and it is also difficult to retrieve the "take-home message". I strongly suggest to reduce where possible the detailed description of specific findings while adding a paragraph for each section (at the beginning or the end of the section) in which the main concepts are stated. This is already partially present, but it should be expanded. The reader should know that the specific details are provided, but that general considerations are available if in depth reading is not necessary for the reader's purpose.

3) the "Insights" section should be divided, being in the present form too heterogeneous in its content. Please, consider here the same comment at 2).

4) Too many acronyms are used in the text. Those that are not strictly necessary should be avoided (e.g. DP for depotentiation at page 7)

Minor issues:

page 2 line 10: "the mechanisms of synaptic communication, plasticity and electrophysiology": it is not clear what the author means with electrophysiology in this statement.

page 2 line 36-37: all the References should be indicated in the same style (using the corresponding number in this case)

page 5 line 38: iCa2+ should be defined

page 6 line 39: "and cols" should be "and colleagues"?

page 7 line 25: the fact that a flux-independent preNMDAR mechanism might be involved is not evident from the previous sentence. APV could act by blocking ion flux in the presynaptic receptor. This conclusion should be shifted to the next sentence, that indeed provides the evidence for that.

page 10 line 9: secondary Ca2+ influx is abbreviated as "2i". The use of this abbreviation is then difficult to read, and it is often used as 2i alone or as 2i Ca2+. The author might consider to change it.

page 16 line 39: "that inhabit in EC milieu": it is not clear what the author is stating... maybe consider rephrasing.

The expression "exciting but thrilling" is repeated in the text, but they are mostly synonyms.

Many typos and errors are present in the text. Just as examples:

page 2 line 17: NMDAR's should be NMDARs (this same typo is repeated at other lines)

page 5 line 3: in the title Nmdar should be NMDAR

page 5 line 28: returned is a typo

page 7 line 34: "mediates" should be "mediate"

page 8 line 39: "mediate" should be "mediated"

page 9 line 34: Hek should be HEK

page 11 line 30: "blocked" instead of "block"; same line, "reduced by 26%".

page 11 line 37: "NMDAR" should probably be "NMDA" here

Author Response

Major issues:

1) English should be revised.

This has been done.

2) The main concern is that the review is mostly a summary of different papers, often merely reporting results in details, and adding the authors' conclusions. In this way, the review is difficult to read and it is also difficult to retrieve the "take-home message". I strongly suggest to reduce where possible the detailed description of specific findings while adding a paragraph for each section (at the beginning or the end of the section) in which the main concepts are stated. This is already partially present, but it should be expanded. The reader should know that the specific details are provided, but that general considerations are available if in depth reading is not necessary for the reader's purpose.

Indeed, several details have been included in the text with the goal to bring attention in these methodological or experimental details that could be relevant for the fiNMDARs cell biology, that today presents several controversies or incongruities. In my experience on cell biology with astrocytes, and that of others, the oversight of these details results into a slower advancement of knowledge and apparent contradictions that take longer to solve. Thus, given the complexity of the NMDAR the highlight of these details could represent and advantage for further work from different labs.

The take home message for each section has been expanded and made clearer where possible.

3) the "Insights" section should be divided, being in the present form too heterogeneous in its content. Please, consider here the same comment at 2).

This section has been separated into subsections and take home messages have been extended and made clearer where possible.

4) Too many acronyms are used in the text. Those that are not strictly necessary should be avoided (e.g. DP for depotentiation at page 7).

This has been corrected

Minor issues:

page 2 line 10: "the mechanisms of synaptic communication, plasticity and electrophysiology": it is not clear what the author means with electrophysiology in this statement.

Most of NMDAR research has been performed in the neuronal-synaptic context. Nevertheless, the NMDAR is widely expressed in different cells and tissues, in which its function has been poorly studied. Therefore, the fiNMDARs becomes also a matter of cellular and molecular biology.  Thus, my intention with this review is to establish a perspective that goes beyond the neuronal, synaptic, plasticity and electrophysiological (the main approach to study NMDAR in neurons) framework. This is also because my scientific experience mainly relates with cellular and molecular biology, although I did a M.Sc. in neurobiology. In an effort to make this point clearer, in the manuscript the word “neuronal” has been added to “electrophysiology”.  

page 2 line 36-37: all the References should be indicated in the same style (using the corresponding number in this case)

This has been corrected

page 5 line 38: iCa2+ should be defined

This has been corrected and defined in the Essentials section where it is used for the first time.

page 6 line 39: "and cols" should be "and colleagues"?

This has been corrected.

page 7 line 25: the fact that a flux-independent preNMDAR mechanism might be involved is not evident from the previous sentence. APV could act by blocking ion flux in the presynaptic receptor. This conclusion should be shifted to the next sentence, that indeed provides the evidence for that.

This has been corrected.

page 10 line 9: secondary Ca2+ influx is abbreviated as "2i". The use of this abbreviation is then difficult to read, and it is often used as 2i alone or as 2i Ca2+. The author might consider to change it.

The 2i abbreviation for “secondary Ca2+ influx” has been changed to “Ca2+ 2i”. The 2i was maintained because it was used in the original paper.

page 16 line 39: "that inhabit in EC milieu": it is not clear what the author is stating... maybe consider rephrasing.

This has been rephrased.

The expression "exciting but thrilling" is repeated in the text, but they are mostly synonyms.

This has been corrected.

Many typos and errors are present in the text. Just as examples:

page 2 line 17: NMDAR's should be NMDARs (this same typo is repeated at other lines)

This has been corrected.

page 5 line 3: in the title Nmdar should be NMDAR

This has been corrected.

page 5 line 28: returned is a typo

This has been corrected.

page 7 line 34: "mediates" should be "mediate"

This has been corrected.

page 8 line 39: "mediate" should be "mediated"

This has been corrected.

page 9 line 34: Hek should be HEK

This has been corrected.

page 11 line 30: "blocked" instead of "block"; This has been corrected.

 same line, "reduced by 26%" This is correct.

page 11 line 37: "NMDAR" should probably be "NMDA" here

This is correct

Reviewer 3 Report

Starting from the introduction, the authors claimed that the aim of this review was to put mechanisms and pathways involved in flux-independent NMDAR signalling (fi-NMDARs) under the attention of cell biologists, rather than focusing on mechanisms of synaptic communication, plasticity and electrophysiology which have been addressed in other published reviews.

However, a long space in the manuscript has been occupied by the paragraph ‘Neuronal Death and Survival’, neuronal fates, which for sure both rely on synaptic communication. Moreover, one of the first observations on the fi-NMDARs consists on its sufficiency in inducing synaptic long-term depression (LTD). The bibliography needs also to be updated.

Indeed, authors should:

a) better address the other neuronal functions in which fi-NMDARs has been found to be involved or alternatively 

b) shorten dramatically the discussion on the role of fi-NMDARs in neuronal death and survival;

c) cite the recent review ‘Gray JA, Zito K and Hell JW. Non-ionotropic signaling by the NMDA receptor: controversy and opportunity [version 1; referees: 2 approved] F1000Research 2016, 5(F1000 Faculty Rev):1010 (doi: 10.12688/f1000research.8366.1);

d) update citations on the Glu concentrations reached in the synaptic cleft. The authors clearly underlined that high Glu concentration are used to elicit a non ionotropic NMDAR response. Accordingly, more recent findings on the evidence that the Glu concentrations may reach the mM range at the synaptic cleft should be reported to demonstrate the physiological relevace of the fi-NMDARs, al least in the nervous system.

Author Response

Starting from the introduction, the authors claimed that the aim of this review was to put mechanisms and pathways involved in flux-independent NMDAR signalling (fi-NMDARs) under the attention of cell biologists, rather than focusing on mechanisms of synaptic communication, plasticity and electrophysiology which have been addressed in other published reviews.

Most of NMDAR research has been performed in the neuronal-synaptic context. Nevertheless, the NMDAR is widely expressed in different cells and tissues, in which its function has been poorly studied. Therefore, the fiNMDARs becomes also a matter of cellular and molecular biology.  Thus, my intention with this review is to establish a perspective that goes beyond the neuronal, synaptic, plasticity and electrophysiological (the main approach to study NMDAR in neurons) framework, rather than to attract the attention of cell biologists as it could have been understood. This is also because my scientific experience mainly relates with cellular and molecular biology, although I did my M.Sc. in neurobiology.

However, a long space in the manuscript has been occupied by the paragraph ‘Neuronal Death and Survival’, neuronal fates, which for sure both rely on synaptic communication. Moreover, one of the first observations on the fi-NMDARs consists on its sufficiency in inducing synaptic long-term depression (LTD).

The space occupied by each section is dictated by the number of references of aech topic and their complexity. Despite the idea is to put this information within the context of cell biology, since most work has been done in neurons, sometimes it may be difficult to go beyond the synaptic communication framework. For these reasons and that mentioned in your comment (the first observation of f-iNMDARs in LTD) the LTD section is the longest in the review.

The bibliography needs also to be updated.

This has been done according to your comments.

Indeed, authors should:

a) better address the other neuronal functions in which fi-NMDARs has been found to be involved or alternatively 

b) shorten dramatically the discussion on the role of fi-NMDARs in neuronal death and survival;

As I mentioned above, the length of each section is dictated by the number of references on the subject, their complexity and the amount of novel information provided regarding f-iNMDARs, not by a biased purpose regarding the topics. The only particular goal of this author was to interpret them form the cell and molecular biology, my main experience, for which neuronal death and survival is relevant, although it is true that synaptic communication is one of the variables involved in it, although not the only one. In an effort to perform an in-depth and objective review of the references this author analyzed and synthesized them in an unbiased manner. However, if the reviewer feels that some specific topic or work is underrepresented, that has not been tackled in other published reviews, do please be more specific to perform the appropriate adaptations.

c) cite the recent review ‘Gray JA, Zito K and Hell JW. Non-ionotropic signaling by the NMDA receptor: controversy and opportunity [version 1; referees: 2 approved] F1000Research 2016, 5(F1000 Faculty Rev):1010 (doi: 10.12688/f1000research.8366.1);

Thank you for the reference, indeed it was not included for mistake, now it is cited in the manuscript.

d) update citations on the Glu concentrations reached in the synaptic cleft.

After searching for a new reference tackling this issue, I was unable to find a more recent reference of it. Indeed, a recent excellent review by Hansen et al. (JGP-Vol 150; no.8: 1081-1105) and other works modelling synaptic biophysics, also cite the same reference by Clements 1992 for this purpose. If you are aware of a novel work on this subject I kindly ask you to please let me know to include it in the manuscript.

The authors clearly underlined that high Glu concentration are used to elicit a non ionotropic NMDAR response. Accordingly, more recent findings on the evidence that the Glu concentrations may reach the mM range at the synaptic cleft should be reported to demonstrate the physiological relevace of the fi-NMDARs, al least in the nervous system.

I apologize for this confusion, but in the “Insights” section it is stated (pag. 15 line 48-50): “An intriguing aspect regarding f-iNMDARs is the wide range of Glu or NMDA concentrations that have been employed to unmask f-iNMDARs. This range encompass from mM to mM concentrations.” Therefore, please do let me know how this confusion arose to make the proper changes to the text.